# ASPPs multimerize protein phosphatase 1

**Derek T. Wei**[1,2], **Kayleigh N. Morrison**[1,3], **Gwendolyn M. Beacham**[1,2¤a],
**Erika Beyrent**[1,2¤b], **Cyrus A. Habas**[1,2], **Ying Zhang**[4], **Laurence Florens**[4],
**Gunther Hollopeter**[1,2,3]*

**1** Department of Molecular Medicine, Cornell University, Ithaca, New York, United States of America,
**2** Field of Biochemistry, Molecular, and Cell Biology, Cornell University, Ithaca, New York, United States of
America, **3** Field of Biomedical and Biological Science, Cornell University, Ithaca, New York, United States
of America, **4** Stowers Institute for Medical Research, Kansas City, Missouri, United States of America

¤a Current address: Department of Medicine, Section of Hematology and Medical Oncology, Boston
University Chobanian and Avedisian School of Medicine and Boston Medical Center, Boston,
Massachusetts, United States of America.
¤b Current address: Penn State Neuroscience Institute – University Park, The Pennsylvania State
University and Department of Engineering Science and Mechanics, The Pennsylvania State University,
University Park, Pennsylvania, United States of America.
* gh383@cornell.edu

org/10.1371/journal.pgen.1011731

Madison, UNITED STATES OF AMERICA

**Peer Review History:** PLOS recognizes the
benefits of transparency in the peer review
process; therefore, we enable the publication
of all of the content of peer review and
author responses alongside final, published
articles. The editorial history of this article is
available here: https://doi.org/10.1371/journal.
pgen.1011731

## Abstract

Protein Phosphatase 1 (PP1) activity is thought to be spatiotemporally defined by
hundreds of different regulatory subunits, but their mechanisms of action are largely
unknown. The Ankyrin repeat, SH3-domain, and Proline-rich region containing Pro-
teins (ASPPs) bind and localize PP1 to cell-cell junctions. Here, we show ASPPs bind
superstoichiometric amounts of PP1. Missense mutations in the ankyrin repeats of
ASPPs, that were previously isolated from a forward genetic screen in *Caenorhab-
ditis elegans*, reduce the stoichiometry of PP1 binding. Forcing PP1 oligomerization
restores mutant ASPP function *in vivo*. We propose that ASPPs multimerize PP1 to
establish a concentrated hub of phosphatase activity at cell-cell junctions.

## Author summary

We have elucidated a new mechanism governing protein phosphatase 1 (PP1)
activity. A family of proteins called the ASPPs function to spatially regulate PP1
by recruiting active phosphatase to specific subcellular locations. Critically, we
observed that ASPPs promote the formation of higher-order PP1 assemblies – a
previously unrecognized regulatory mechanism. We identified specific ASPP mu-
tants in our nematode model organism that disrupt PP1 oligomerization, leading
to altered development. However, inducing PP1 clustering was sufficient to rescue
these ASPP mutants, underscoring the functional significance of ASPP-mediated
PP1 oligomerization. These results provide new insights into the intricate control of
cellular signaling pathways mediated by PP1 and may have implications for under-
standing diseases associated with dysregulated phosphatase activity.

**Data availability statement:** All underlying numerical data for all graphs and summary statistics are publicly available using eCommons@Cornell via doi.org/10.7298/DW6T-E066. Raw TIRF microscopy images are available at BioImage Archive via doi.org/10.6019/S-BIAD2337. Mass spectrometry data may be accessed through Proteome Xchange (accession: PXD065626) at the MassIVE repository via ftp://massive-ftp.ucsd.edu/v10/MSV000098385/. Original mass spectrometry data underlying this manuscript may also be accessed from the Stowers Original Data Repository via stowers.org/research/publications/LIBPB-2566.

**Funding:** GH is funded by NIH R01 GM127548 and R35 GM156633 from the National Institutes of Health. https://www.nih.gov/ The funder played no role in the study design, data collection and analysis, decision to publish, or preparation of the manuscript.

**Competing interests:** The authors have declared that no competing interests exit.

## Introduction

There are far fewer protein phosphatases than protein kinases in the human genome [1,2], so how dephosphorylation reactions are sculpted in order to match kinase diversity is an open question. The predominant serine/threonine protein phosphatase catalytic subunit, Protein Phosphatase 1 (PP1), is thought to achieve spatiotemporal precision by binding hundreds of different regulatory subunits [3,4]. These regulatory subunits have so far been shown to specify subcellular localization and substrate selection of the catalytic subunit. For example, a glycogen-binding regulatory subunit is thought to bring the PP1 catalytic subunit to glycogen in order to focus phosphatase activity on nearby metabolic enzymes [5]. Another example is spinophilin, which directs PP1 to act on glutamate receptor 1 while preventing reactions with other substrates, such as phosphorylase a [6]. However, mechanisms of action for most regulatory subunits are unknown [7].

The Ankyrin repeat, SH3-domain, and Proline-rich region containing Proteins (ASPPs) are a family of medically important PP1 regulatory subunits with an unclear function. There are three vertebrate ASPP homologs – ASPP1, ASPP2, and iASPP. Forward-genetic studies across cows [8–10], mice [11,12], and humans [13,14] connect mutations in iASPP, the most conserved homolog [15], to a lethal cardiocutaneous disease. While studies originally reported that the ASPPs interact with p53 [16–18], it has been difficult to attribute p53 dysregulation to phenotypes associated with loss of iASPP [19]. The ASPP C-terminal ankyrin repeats and SH3 domain bind PP1 with an affinity that is ~100-fold higher than p53 [20–25]. Additionally, mutating ASPP residues predicted to interact with PP1 produces a phenotype similar to the ASPP null mutant in *Drosophila* [24]. These data suggest that ASPPs may regulate PP1, but the mechanism is unclear.

Several groups have proposed that ASPPs function at cellular junctions. iASPP colocalizes with desmosomes in cardiomyocytes and keratinocytes [19,26]. iASPP knockout and knockdown also results in reduced expression of numerous junctional components that are found in desmosomes, tight junctions, GAP junctions, and adherens junctions [19,27,28]. ASPP2 also localizes to apical cell-cell junctions in epithelial cells and associates with the PAR polarity complex [29–31]. Deletion of ASPP2 in mice reduces PAR3 localization to apical cell junctions and slows the formation of tight junctions [30,32]. ASPP2 null mice are similar to iASPP mutants in that they are postnatally lethal and have skin and heart defects [29,33–35]. In *Drosophila melanogaster*, the sole ASPP homolog is also reported to colocalize with adherens junction components [36] and its knockout leads to disorganized epithelial cell junctions [37]. Therefore, ASPPs appear to have a conserved role in regulating cellular junctions.

One holistic model proposed for the ASPPs is that they bind and localize PP1 to cell junctions in order to dephosphorylate specific junctional components [24,32,38,39]. Consistent with this model, we found that phosphatase localization to epithelial cell-cell junctions is dependent on the sole ASPP homolog in *C. elegans* [25]. However, we demonstrated that mere localization of phosphatase to epithelial junctions could not recapitulate ASPP function [25]. Instead, we proposed that the

highly conserved ASPP C-terminal ankyrin repeats and SH3 domain (hereafter called the ASPP 'C-terminus') is required to modulate PP1 activity through an additional unknown mechanism.

In this study, we find that ASPPs perform a new, unanticipated function – clustering of PP1. In addition to localizing and promoting PP1 activity at epithelial junctions, we discover that ASPPs bind superstoichiometric amounts of PP1. We show that semi-dominant ASPP mutations, identified in a *C. elegans* suppression screen, reduce the ratio of PP1 to ASPP. Forcing oligomerization of the phosphatase catalytic subunit restores ASPP function in these mutants. We therefore propose that ASPPs multimerize PP1 to form dephosphorylation centers at epithelial cell junctions.

## Results

### APE-1 function requires an N-terminal helix that localizes to epithelial junctions

Previously, we found that the N-terminal 519 residues of the sole *C. elegans* ASPP homolog (APE-1) are required for both function and localization [25]. However, the protein sequence mediating localization was unclear as this region is poorly conserved. Using AlphaFold structural prediction, we noted that a region predicted to form a helix of ~100 residues is embedded within the first half of APE-1 (Figs 1A, S1A). We truncated the endogenous APE-1 using CRISPR to see if this region might encode a localization signal (Fig 1A). We found that the presumptive helix does localize a GFP tag to epithelial junctions, marked by MLT-4::RFP (Fig 1B).

We then assessed which pieces of APE-1 are needed for function. Loss of APE-1 suppresses jowls – overt cyst-like bulges near the head of the animal caused by hyperactivation of the Inversin homolog, MLT-4 [40]. Our hyperactive alleles of MLT-4 include a C-terminal red fluorescent protein (RFP) tag or a missense mutation (E470K). The MLT-4::RFP allele also results in reduced body length. Therefore, we can assay for APE-1 function by quantifying the percentage of animals with jowls or measuring their body lengths. We found that neither APE-1's N-terminal helix nor the C-terminus is functional on its own, according to our jowls and body length assays (Figs 1C, S1B). We also noted that MLT-4::RFP intensity was reduced in these APE-1 truncations (S1C Fig), suggesting APE-1 activity may also facilitate MLT-4 localization. Fusing the N-terminal helix to APE-1's C-terminal domains creates a mini-construct that not only localizes to epithelial junctions (Fig 1B), but is also functional (Figs 1C, S1B). Therefore, APE-1 function likely requires its N-terminal helix to localize the APE-1 C-terminus, with bound phosphatase, to epithelial junctions. These results suggest that APE-1 confers subcellular localization to PP1, one of the canonical roles proposed for phosphatase regulatory subunits.

### ASPPs promote PP1 activity

PP1 regulatory subunits are also thought to control the activity state of the PP1 catalytic subunits. We and others have shown that ASPPs bind PP1 through the highly conserved ASPP C-terminus [20,24,41]. While we proposed that the ASPP C-terminus is required for modulating PP1, it was unclear if this modulation promoted or inhibited PP1 activity [25].

We hypothesized that if ASPPs stimulate PP1 activity, then deletion of the *C. elegans* PP1 homolog (GSP-2) would suppress jowls. However, since null alleles of GSP-2 are lethal, we degraded GSP-2 specifically in the *C. elegans* hyp7 epithelium using the auxin-inducible degron system (Fig 2A). We found that hyp7-specific degradation of GSP-2 suppressed jowls in our two mutants (Fig 2A). Therefore, GSP-2 is required in the epithelium for APE-1 function. These data are consistent with ASPPs promoting PP1 activity at epithelial junctions.

PP1 is thought to be inactivated by a phosphorylation event at T320 [42,43]. To test if our model that ASPPs promote PP1 activity is correct, we queried whether ASPPs bind PP1 that is dephosphorylated (active). We transfected HEK293T cells with a plasmid of the C-teriminus of iASPP – the most conserved mammalian ASPP homolog [15] – fused to a HaloTag. We isolated the iASPP C-terminus and any bound PP1 via HaloTag purification. We found that only dephosphorylated (active) PP1 co-purified with the iASPP C-terminus (Fig 2B). This result is consistent with ASPPs recruiting active PP1. Thus, ASPPs specify both the localization and activity state of PP1, consistent with the two modes of action posited for regulatory subunits [4].

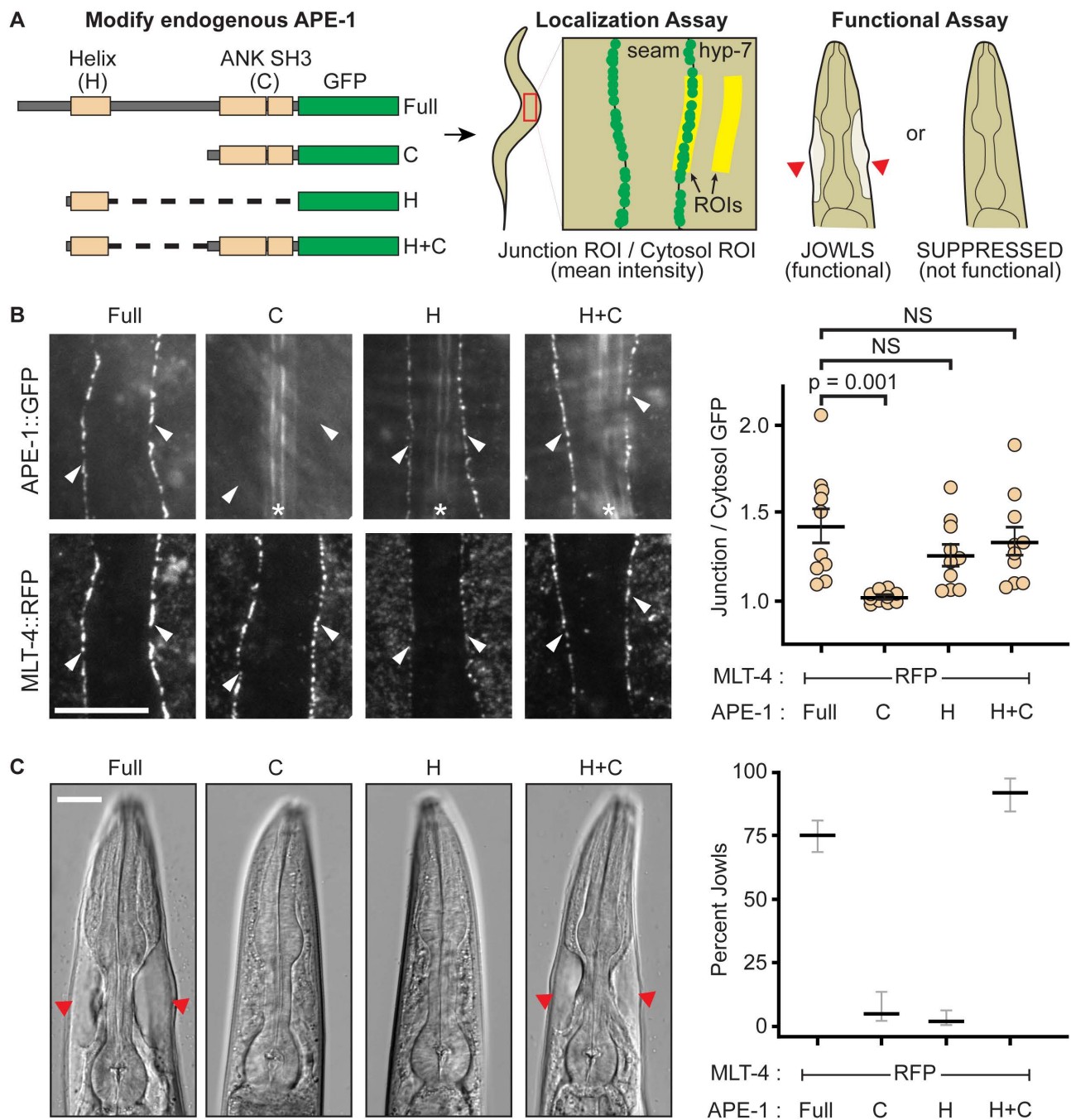

**Fig 1. APE-1 function requires its N-terminal helix. (A)** Experimental design. Endogenous APE-1 was truncated and GFP-tagged via CRISPR. APE-1 constructs were then assessed for localization to epithelial junctions between the seam and hyp-7 cell syncitia via TIRF microscopy. A region of interest (ROI) was drawn over cellular junctions using MLT-4::RFP as a marker. This ROI was duplicated and moved to a nearby hyp-7 cytosolic region. The mean GFP intensity was then calculated for both ROIs and reported as a ratio of junctional intensity over cytosolic intensity. APE-1 constructs were also assayed for function via a jowls assay. **(B)** Localization assay. White arrowheads indicate seam and hyp-7 junctions. * indicates autofluorescent alae. Data represent the mean and S.E.M. (black bars) of 10 biological replicates. Scale bar = 10 μm. NS = not significant. **(C)** Jowls assay. Red arrow heads indicate jowls. Data represent the percentage and 95% confidence interval (black and gray bars) of jowls in populations of adult animals (n = 247, 76, 142, and 164, respectively). Scale bar = 15 μm.

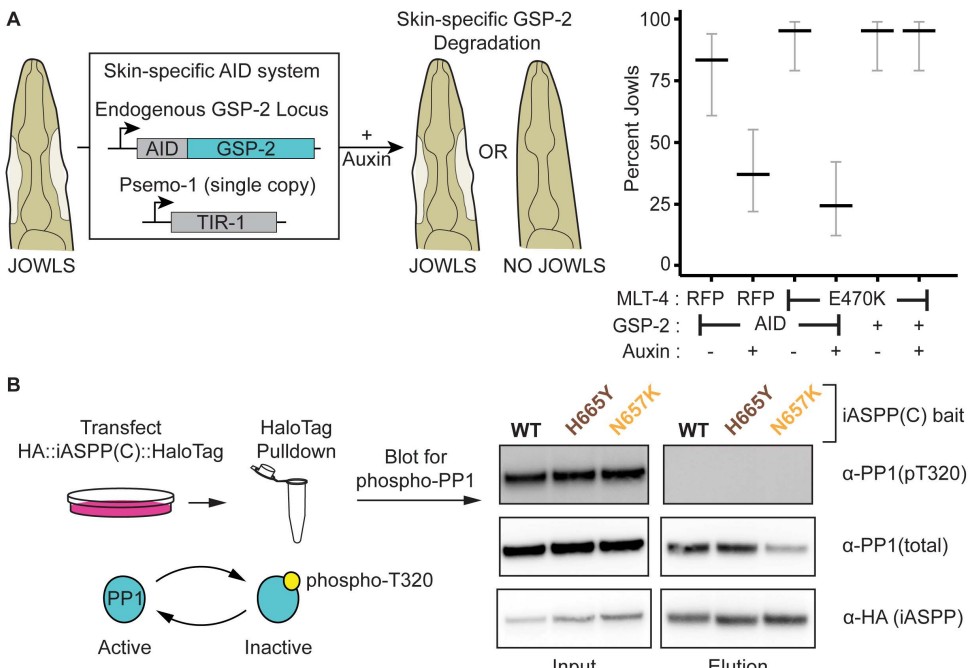

**Fig 2. ASPPs promote PP1 activity. (A)** Skin-specific depletion of GSP-2. Endogenous GSP-2, tagged with an Auxin Inducible Degron (AID), is ubiquitinated by skin-specific TIR-1 when animals are fed auxin. Data represent the percentage and 95% confidence interval (black and gray bars, respectively) of jowls in adult animals (n = 30 each). **(B)** HEK293T cells were transfected with plasmids encoding HaloTagged iASPP C-terminal constructs. Input and elutions were blotted for PP1 phosphorylated at T320, total PP1, and an HA tag on the iASPP bait. Missense mutations (brown and orange) identified from a forward genetic screen, which each inhibit APE-1 in *C. elegans*, were also included in the iASPP bait.

## ASPPs multimerize PP1

We previously showed that localizing PP1 to epithelial junctions is not sufficient to recapitulate ASPP function in *C. elegans* [25]. Instead, the ASPP C-terminus appears to be required to exert some additional effect on PP1. We had identified missense mutations in two highly conserved residues (N583 and H591) of the ASPP ankyrin repeats that each independently render APE-1 inactive [25]. Yet, each mutant is still able to bind and localize PP1, implying that these residues somehow inactivate a key aspect of PP1 regulation. One potential model is that these missense mutations disrupt ASPP selectivity, allowing for the recruitment of phosphorylated (inactive) PP1. To test this model, we included each homologous missense mutation in the iASPP C-terminus (N657K and H665Y) and performed HaloTag purifications from HEK293T cells as described above. We found no change in the phosphorylation state of copurified PP1 with each mutation (Fig 2B). These data suggest the missense mutations alter some other regulatory function of the ASPPs.

Interestingly, while identifying ASPP interactors using semi-quantitative MudPIT proteomics, we noted that both APE-1 and the three vertebrate homologs all co-purified superstoichiometric amounts of PP1 from *C. elegans* (Fig 3A) and HEK293T cell lysates, respectively [25]. Furthermore, introducing either missense mutation identified in our genetic screen reduced the amount of GSP-2 copurified with APE-1 (Fig 3A). Since the residues in the ankyrin repeats that seem to control phosphatase oligomerization are highly conserved, we asked whether the analogous mutations would also reduce the stoichiometry of PP1 bound to iASPP, the most conserved vertebrate homolog. We cotransfected HEK293T cells with plasmids to express PP1 fused to an HA tag and the iASPP C-terminus fused to a HaloTag and HA tag. We performed HaloTag purifications and quantitatively blotted for the HA tags on both iASPP and PP1. We found that the iASPP C-terminus was sufficient to co-purify superstoichiometric amounts of PP1 (Fig 3B). Introducing either of the homologous

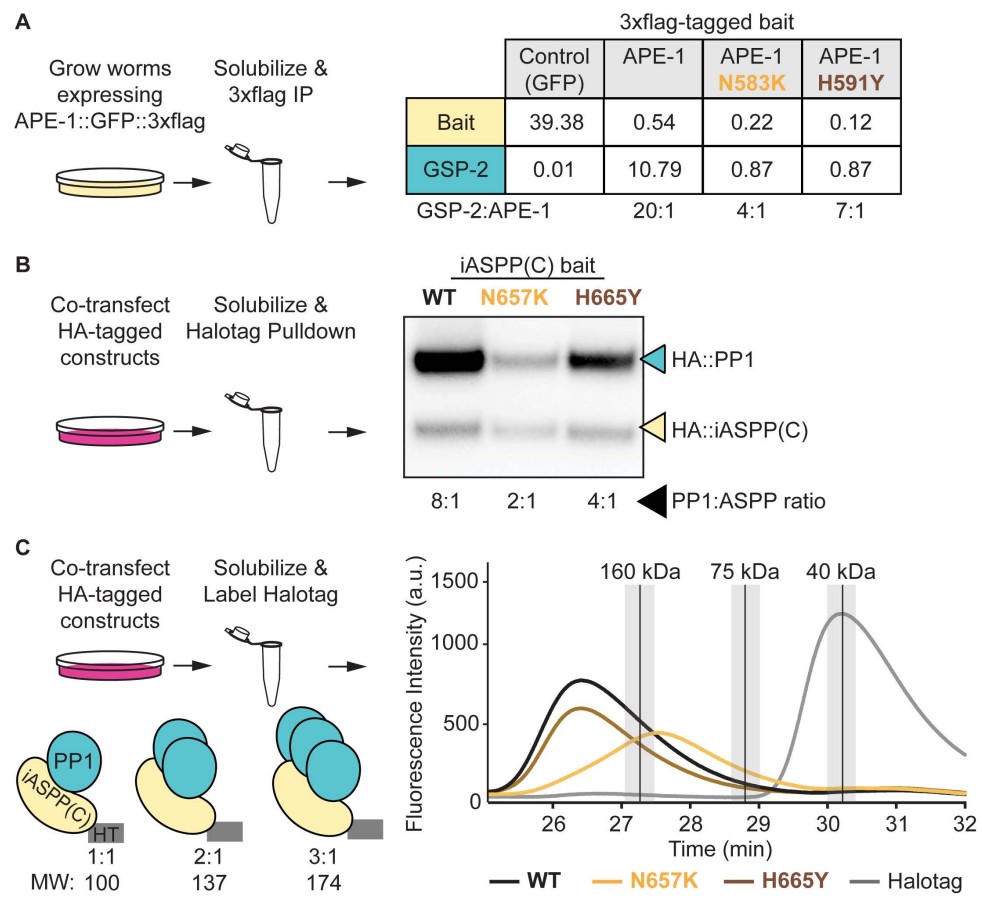

**Fig 3. ASPPs multimerize protein phosphatase 1. (A)** Multidimensional Protein Identification Technology (MudPIT) analysis of 3xflag immunoprecip-itations from whole-worm lysates. Data represent the amount of bait and GSP-2 as a percentage of total protein found in each sample. Ratios of GSP-2 to APE-1 are shown below the table. Note that the control and APE-1 MudPIT data are from [25]. **(B)** Quantitative HA blot of HaloTag purifications from HEK293T cells. Ratios of PP1 to iASPP C-terminus are shown below the blot. **(C)** Fluorescent size exclusion chromatography of HaloTagged (HT) proteins labeled with JFX549 in HEK293T cell lysates. Peaks for molecular weight standards (described in methods) are represented as vertical black lines with the top 10% of signal shaded in light gray. PP1 is 38.8 kDa and iASPP(C)::HT is 62.1 kDa. The HaloTag negative control protein (gray line) is approximately 37.2 kDa.

mutations identified in our genetic screen reduced the ratio of PP1 to iASPP (Fig 3B). Therefore, both *Homo sapiens* iASPP and *C. elegans* APE-1 appear to form a complex with superstoichiometric amounts of PP1 that depends on two key residues in the ASPP ankyrin repeats.

To approximate the size of the ASPP:PP1 complex, we solubilized HEK293T cells co-transfected as described above and covalently labeled the HaloTagged iASPP C-terminal constructs with the fluorescent dye JFX549. We performed fluorescent size exclusion chromatography (fSEC) on these labeled samples by monitoring for JFX549 fluorescence in the elutions. We compared the elution times of fluorescent ASPP:PP1 complexes with protein standards of known molecular weights. The wild type complex eluted earlier than the 160 kDa standard, (Fig 3C) – consistent with the iASPP C-terminus complexing with superstoichiometric amounts of PP1. Introducing one of the homologous missense mutations, N657K, into the iASPP C-terminus shifted the complex elution time to between the 75 kDa and 160 kDa standards (Fig 3C) – con-sistent with a reduction in the amount of PP1 bound to the iASPP C-terminus. A 1:1 ratio would be ~100 kDa. Interest-ingly, introducing the second homologous missense mutation, H665Y, into the iASPP C-terminus did not overtly shift the

complex's size (Fig 3C). These data suggest the H665Y mutation is weaker at disrupting the ASPP:PP1 complex than the N657K mutation. Indeed, the H665Y missense mutation reduces the amount of PP1 in the ASPP:PP1 complex to a lesser extent than the N657K missense mutation in both the semi-quantitative proteomics (Fig 3A) and quantitative blotting (Fig 3B). Together, these data suggest ASPPs multimerize PP1 and that the ASPP ankyrin repeats assist in phosphatase oligomerization.

We next sought additional evidence for PP1 oligomers. We found that a single-copy, skin-specific, 3xflag::GFP::GSP-2 bait coprecipitated with endogenous HaloTag::GSP-2 (S2A Fig). Knockout of APE-1 reduced the amount of HaloTag::GSP-2 coprecipitation, but not to the level of the 3xflag::GFP negative control bait. Perhaps other proteins cooperate with the ASPPs to promote phosphatase oligomerization. For instance, CCDC85 and RASSF8 are ASPP interactors hypothesized to regulate cellular junctions via PP1 in *Drosophila melanogaster* [24,36]. Given that two independent deletion alleles for each CCDC85 and RASSF8 failed to suppress the jowls phenotype (S2B Fig), it appears that neither are involved in ASPP oligomerization of PP1 in our system. It is also possible that the residual oligomerized phosphatase in our precipitations may represent other, ASPP-independent, complexes. If the ASPP-dependent oligomerization of PP1 requires no other eukaryotic proteins, then it may be possible to reconstitute the complex using components purified from bacteria. We mixed recombinantly purified iASPP C-terminus and PP1 at a 1:3 molar ratio, respectively (S2C Fig). Using size exclusion chromatography, we observed that the ASPP:PP1 complex eluted at a size consistent with a 1:1 stoichiometry, which is reminiscent of the iASPP(602–828, N657K) complex in the fSEC data (Fig 3C). The lack of evidence of oligomerized PP1 using recombinant proteins suggests that we are missing some key factor for oligomerization. It has been established that PP1 purified from bacteria has non-native metal ions in its active site and exhibits inappropriate activity [44], which may account for our inability to reconstitute the superstoichiometric complex. However, we cannot exclude the possibility that PP1 oligomerization depends on additional proteins working with the ASPPs.

### Single-molecule analysis shows superstoichiometric ASPP:PP1 complexes

We next evaluated ASPP:PP1 complexes at the single-molecule level using Total Internal Reflection Fluorescence (TIRF) microscopy (Fig 4A). We isolated endogenous APE-1, tagged with 3xflag and GFP, from *C. elegans* whole-worm lysates using coverslips functionalized with anti-flag antibody. These animals also had endogenously HaloTagged GSP-2 labeled with the far-red dye JF646. We identified spots where APE-1 (green) colocalized with GSP-2 (far-red). We reasoned that if ASPP can bind more than one PP1, then, under continuous excitation, green spots exhibiting one photobleaching event should colocalize with far-red spots exhibiting multiple photobleaching events. We trained two Convolutional Neural Networks (CNNs) to classify stepwise photobleaching events in both green and far-red fluorescence intensity traces. Our "CNN-Green" and "CNN-FarRed" showed mean validation accuracies of 85.6 +/- 1.2% and 73.2 +/- 1.0%, respectively, during 5-fold cross validations with our pre-labeled training data (S3 Fig, S1 and S2 Files). Using our trained CNNs, we found that 39% of ASPP:PP1 complexes, which contained a single APE-1, colocalized with multiple GSP-2 molecules (Fig 4B). Introducing the N583K missense mutation into APE-1 resulted in a reduction in the amount of complexes containing oligomeric GSP-2 (Fig 4B, p=0.008). The H591Y missense mutation in APE-1 did not significantly alter the stoichiometry of the complex, which is similar to our other *in vitro* assays where this mutation had less of an effect on the ASPP:PP1 complex. To evaluate if the missense mutations stabilize a specific GSP-2 oligomeric state, we segregated the data to quantify the distribution of GSP-2 monomers, dimers, and trimers-or-higher (S3E Fig). By this analysis, there is no statistically significant difference between the wild type and the missense mutants in the higher order complexes. Our single-molecule analysis is highly dependent on efficiently labeling the HaloTag::GSP-2 with fluorescent dye. Since our labeling efficiency is approximately 40% (S3F Fig), we are likely underestimating the number of phosphatase molecules, and may fail to detect higher order ASPP:PP1 complexes. Regardless, our single-molecule analysis of the ASPP:PP1 complex supports the model that ASPPs can bind multiple PP1 catalytic subunits and that the ASPP ankyrin repeats promote this oligomerization.

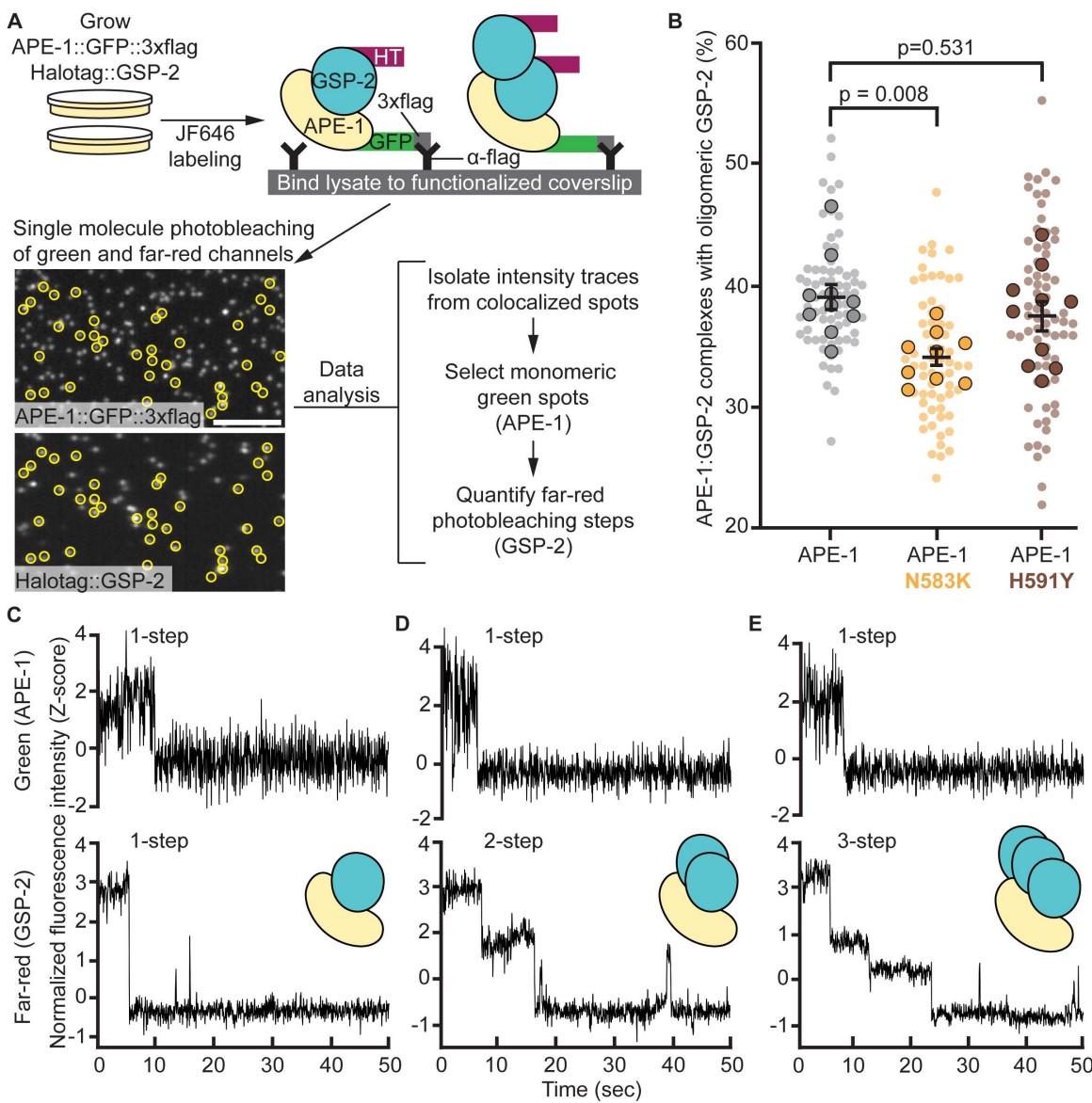

**Fig 4. Single-molecule analysis shows superstoichiometric ASPP:PP1 complexes. (A)** Experimental design for single-molecule analysis of APE-1:GSP-2 complex stoichiometry. Worms with wild type or mutant APE-1 were fed far-red fluorescent dye (JF646) to label HaloTag::GSP-2. Whole worm lysates were loaded on coverslips functionalized with anti-flag antibody. Coverslips were imaged on a TIRF microscope in the green and far-red channels. Fluorescence intensity traces from colocalized green and far-red spots (yellow circles) were isolated for analysis. A convolutional neural network (CNN), trained to recognize single-step photobleaching events in green intensity traces, selected protein complexes containing monomeric APE-1. A second CNN, trained to recognize single and multistep photobleaching events in far-red intensity traces, quantified the number of monomeric APE-1 molecules colocalized with mulitple GSP-2. Scale bar = 5 µm. **(B)** Quantification of APE-1:GSP-2 complexes containing monomeric APE-1 and oligomeric GSP-2. Data represent the mean and S.E.M. (black bars) of 9-10 biological replicates (outlined spots), each consisting of 7 technical replicates (small spots). **(C-E)** Example green (top panels) and far-red (bottom panels) normalized fluorescence intensity (Z-score) traces from colocalized spots.

## Phosphatase oligomers restore function to ASPP missense mutants *in vivo*

If a key function of the ASPP C-terminus is to oligomerize PP1, we hypothesized that forcing GSP-2 oligomerization should restore function to the APE-1 proteins containing the ankyrin repeat missense mutations (Fig 5A). To test this

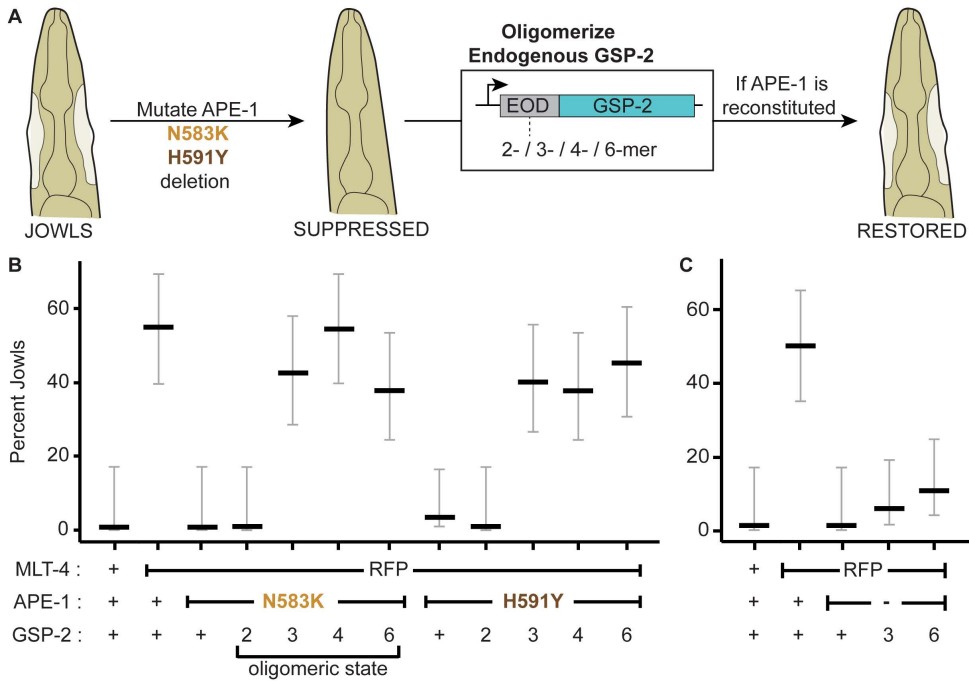

**Fig 5. GSP-2 oligomers bypass APE-1 missense mutations, but not APE-1 deletion. (A)** Extra-oligomerization domains (EODs) favoring different homo-oligomers are attached to the endogenous GSP-2 of animals where jowls had been suppressed by mutations in APE-1. Animals are then assessed for the restoration of jowls. **(B)** Jowls assay for APE-1 missense mutants. **(C)** Jowls assay for APE-1 deletion mutant. Data represent percent jowls and 95% confidence interval (Black and gray bars) in L4 animals (n = 40 each).

hypothesis, we fused extra-oligomerization domains (EODs) to the N-terminus of endogenous GSP-2 in *C. elegans* using CRISPR. EODs are domains from a variety of proteins that have been shown to favor homo-oligomers of various sizes *in vivo* [45]. We selected EODs that would favor a dimeric, trimeric, tetrameric, or hexameric GSP-2. Fusing the dimeric EOD to GSP-2 was not sufficient to restore jowls, but did reduce body length suggesting APE-1 function was not fully recapitulated (Figs 5B, S4A). Instead, fusing a trimeric or higher-order EOD to GSP-2 was sufficient to reconstitute APE-1 function in the presence of either missense mutation, as indicated by both jowls and body length assays (Figs 5B, S4A). As expected, GSP-2 oligomers failed to bypass the complete deletion of APE-1 (Fig 5C), since the regulatory subunit is also required to recruit and activate the GSP-2 complex at epithelial junctions (Figs 1 and 2). We also tested an alternative EOD, a modified *Arabidopsis thaliana* Cryptochrome 2 (Cry2olig) that drives protein clustering in response to light [46]. Fusing Cry2olig to GSP-2 restored jowls in the N583K, but not in the H591Y missense mutants (S4B Fig), suggesting that these two mutations may inactive the ASPP:PP1 complex via slightly distinct mechanisms. One caveat is that the Cry2olig did not exhibit light sensitivity, but rather temperature sensitivity – at room temperature animals exhibited jowls both in the light and dark, whereas the jowls were suppressed at 15°C (S4C Fig). Altogether, these *in vivo* data complement the *in vitro* data to suggest that, in addition to localizing active PP1, a core function of ASPPs is to multimerize PP1.

## Discussion

Our study refines and expands the model for ASPP regulation of PP1. We found that the *C. elegans* ASPP contains a presumptive N-terminal helix that specifies localization to epithelial junctions (Fig 1) and that ASPPs likely promote PP1 activity (Fig 2). Interestingly, we also discovered that ASPPs oligomerize PP1 via their ankyrin repeats (Figs 3 and 4) and

that we can bypass inactivating mutations in this conserved region by engineering PP1 oligomers in *C. elegans* (Fig 5). Altogether, our data are consistent with a new model whereby ASPPs cluster PP1 activity at epithelial junctions.

AlphaFold predicts that the vertebrate ASPPs also contain N-terminal helices (S1 Fig), yet whether the vertebrate helices specify subcellular localization or are required for function is unclear. Some studies have shown that regions of the ASPP N-termini bind proteins such as desmoplakin [26] and PAR-3 [29,30,38]. These proteins may act as molecular anchors allowing the vertebrate ASPP:PP1 complex to localize to epithelial junctions. We were unable to localize GFP to epithelial cell junctions in *C. elegans* using the predicted vertebrate helices. One possibility is that the binding partners required to localize the vertebrate helices are not present in the worm hypodermis. Alternatively, the binding partners are present, but the interacting residues are not conserved – the N-terminal regions of the ASPPs are poorly conserved. Future studies will be required to further clarify the molecular mechanisms of ASPP localization.

Our data are consistent with ASPPs promoting PP1 activity – epithelial GSP-2 is required for APE-1 function (Fig 2A). However, it is formally possible that the phosphatase is required for a non-enzymatic function. Perhaps oligomerized phosphatase provides a structural feature required for junctional organization that is independent of enzymatic activity. We also showed the iASPP C-terminus specifically co-purifies PP1 that is dephosphorylated at T320 (Fig 2B), which has been considered to be an active form of PP1 [42,43]. However, to validate the model that ASPPs promote PP1 activity, a substrate at epithelial junctions needs to be identified. If ASPPs promote PP1 activity, then we predict that loss of ASPP will result in the accumulation of one or more phosphorylated targets in affected tissues. Identification of phosphorylated proteins enriched in ASPP mutants may reveal candidate substrates for the ASPP:PP1 complex and lead to potential therapeutic interventions for the cardiocutaneous disease using specific kinase inhibitors.

Our *in vitro* (Figs 3 and 4) and *in vivo* (Fig 5) data suggest that ASPPs oligomerize PP1, yet precisely how ASPPs induce PP1 oligomerization and the molecular arrangement of the complex remains unclear. Previously, crystal structures of ASPPs bound to PP1 showed a 1:1 complex [24,47]. These structures could represent an initial seeding intermediate or regulatory state distinct from what we observe. The superstoichiometric ASPP:PP1 complex might consist of an ordered array of PP1. Indeed, a *Drosophila* protein phosphatase named Herzog homo-oligomerizes into fibrillar structures via its prion-like domain [48]. The ASPP:PP1 complex relies on a slightly different mechanism whereby the independent regulatory subunit, ASPP, somehow clusters many PP1 molecules into a superstoichiometric complex. Alternative structural approaches such as cryo-electron microscopy might reveal these higher order states whereas crystallization conditions may favor the heterodimeric state. The ASPPs could also induce phase separation of PP1, in which case a structure may be difficult to resolve.

Our *in vitro* assays report different stoichiometries for the ASPP:PP1 complex. Perhaps the ASPP:PP1 stoichiometry is highly sensitive to the method of purification, which varies between our assays. It is also possible that there is inherent variability in ASPP:PP1 stoichiometry. Perhaps subunit stoichiometry changes depending on the tissue in which the complex is expressed or the developmental state of the organism. We used whole-worm lysates of unsynchronized populations in our MudPIT proteomics and single-molecule analysis of the worm ASPP:PP1 complex. Therefore, these analyses lack tissue specificity and resolution in developmental timing. Similarly, our biochemical assays using HEK293T cells capture a snapshot of ASPP:PP1 behavior in a single cell type. Finally, because our labeling efficiency is only 40% (S3F Fig), our single-molecule TIRF microscopy analysis is likely underestimating the number of GSP-2 in many ASPP:PP1 complexes. Nonetheless, all *in vitro* assays point to a shared conclusion – ASPPs can oligomerize PP1. Our assays also consistently show that including either of two semi-dominant suppressor mutations in the ASPP ankyrin repeats reduces PP1 oligomerization, with one mutation (*C. elegans* N583K/*H. sapiens* N657K) having a greater effect than the other (*C. elegans* H591Y/*H. sapiens* H665Y). These data suggest that even a slight decrease in PP1 oligomerization reduces ASPP function. Our *in vivo* data complement our *in vitro* assays in that fusing a trimeric or higher-order EOD to GSP-2 restores APE-1 function in these mutants, while fusing a dimeric EOD to GSP-2 is less effective (Figs 5B, S4A). Our data are

consistent with the model that APE-1 induces GSP-2 oligomerization beyond some threshold to elicit a functional output. However, stimulating oligomerization of GSP-2 with Cry2olig bypassed the N583K missense mutant, but not the H591Y (S4B Fig). These data suggest there may be an additional level of regulation, perhaps a specific orientation of phosphatase subunits, which the H591Y mutant cannot achieve. It seems likely there are active forms of the ASPP:PP1 complex that remain to be determined.

The purpose of ASPP-induced PP1 oligomerization is unclear. In the case of Herzog, oligomerization is thought to enhance the phosphatase's enzymatic activity [48]. It is possible that ASPP-induced PP1 oligomerization also enhances PP1 activity. Perhaps a target substrate is highly concentrated at epithelial junctions and requires substantial dephosphorylation to reach a specific phospho-state. However, an alternative model that remains consistent with our data is that oligomerized PP1 may function in a non-enzymatic, structural capacity. While the functional significance of the ASPP:PP1 complex remains an open question, our discovery of PP1 oligomerization by ASPPs may represent a general mechanism of PP1 regulation.

## Materials and methods

### Worm strains, maintenance, and CRISPR-Cas9 transgenics

Strains were maintained at room temperature on 60 mm petri dishes containing nematode growth media (NGM) seeded with a bacterial food source (OP50). In Fig 2A, strains were also maintained on petri dishes as described above with 1 mM auxin. CRISPR-Cas9 edits were generated as described in [49] with modifications described in [25]. Alleles were verified by Sanger sequencing of a PCR amplicon of the modified locus. A complete list of components of the RNP complexes, repair strategies, and resulting strains and alleles can be found in S3 File.

### Structural prediction and molecular visualization

Structural predictions of *C. elegans* APE-1 (AF-Q9XVN3-F1-v4) and the *Homo sapiens* ASPP1 (AF-Q96KQ4-F1-v4), ASPP2 (AF-Q13625-F1-v4), and iASPP (AF-Q8WUF5-F1-v4) were acquired from the AlphaFold repository [50,51]. Molecules were visualized in ChimeraX [52,53] and colored according to the reported pLDDT value for each residue.

### DIC and live fluorescent imaging

For DIC imaging, adult worms were mounted on 2% agarose pads in M9 buffer containing 20 mM sodium azide on glass slides. Worms were imaged at room temperature on a Nikon Eclipse Ti-S microscope equipped with a Nikon DS-Qi2 camera using a 40x DIC objective (Nikon, NA = 0.75) within 20 minutes of mounting and overlay of No. 1.5 glass coverslip. Images were taken using the NIS-Elements D 5.41.00 software and processed in Fiji [54].

For live fluorescent imaging, adult animals were mounted on 8% agarose pads in 3 µL 1x PBS containing 1.3% (w/v) 0.10 µm Polybead Microspheres (Polysciences Inc., 00876) on glass slides. We imaged red and green channels using a custom-built RM21 TIRF microscope (MadCity Labs) with a 60x oil immersion objective (Nikon, NA = 1.49), 488 and 552 nm lasers (Coherent OBIS), and an Orca Fusion BT sCMOS camera (Hamamatsu). Images were collected at room temperature using Micro-Manager 2.0. MLT-4::tagRFP::HA was used to identify the focal plane for the hyp7 and seam cell junction. Images were then collected in the red and green channels. For image analysis, the MLT-4 signal was used as a marker for apical junctions. A segmented line (5 pixels wide) was traced along the MLT-4 signal as a region of interest (ROI) in Fiji. The ROI was duplicated and moved into the cytosolic region of the hyp7 cell syncytium. Both ROIs were overlaid onto the images collected in the green channel. The mean intensity along each ROI was quantified and the ratio of junctional to cytosolic mean intensities calculated. MLT-4 signal was analyzed in S1C Fig by duplicating the ROI previously drawn in the red channel and moving it to a cytosolic region. The mean intensity along each ROI was quantified and the ratio of junctional to cytosolic mean intensities calculated.

## Body length assays

Body length assays were conducted as in [25]. Briefly, adult animals were mounted on 2% agarose pads in M9 buffer containing 20% (v/v) sodium azide on glass slides. Samples were imaged at room temperature using the BZ-X810 Keyence microscope system with a 10x objective (Nikon, NA=0.45). Body length was measured by drawing a line through the middle of the worm from anterior to posterior in Fiji.

## Jowls assays

In Fig 1, two plates of synchronized worms for each strain were scored for the presence of jowls using a dissection microscope (Nikon SMZ800N). In Fig 2A, 30 adult worms were scored for each condition except GUN1663 with no auxin, which had 20 worms scored. In S2B Fig, 40 adult worms were scored for jowls. In Fig 5, 40 L4 larvae were scored for each strain. Tagging the endogenous GSP-2 with trimeric, tetrameric, or hexameric EODs made adult animals unhealthy and more difficult to score phenotypically. Therefore, phenotypic scoring was conducted on L4 larvae when jowls were easily observable. In S4B Fig, 25–35 adult animals were scored for jowls. In S4C Fig, 50 eggs from GUN1740 were grown for 3–4 days at room temperature or at 15°C. At each temperature, one population was wrapped in aluminum foil to keep animals in the dark while a second population was grown under ambient light. 29–46 adult animals from each condition were scored for jowls.

## Whole worm immunoprecipitations

In Fig 3A, worm samples were prepared as described in [25]. Briefly, worms were grown at room temperature in an OP50 *E. coli* culture mixed with chicken eggs. Worms were collected and washed in ice-cold H150 buffer (50 mM HEPES pH 7.6 and 150 mM KCl). Worms were resuspended in 25 mM HEPES pH 7.6, 75 mM KCl, and 5% glycerol with protease inhibitors (Roche, A32965, 1 tablet per 10 ml lysis buffer) and frozen in liquid nitrogen as small pellets. Frozen worm samples were lysed using mortar and pestle pre-chilled with liquid nitrogen until no intact worms were visible under a dissection microscope. Lysates were diluted 5-fold with 50 mM HEPES pH 7.6, 150 mM KCl, 10% glycerol, 0.005% IGEPAL (CA-630, Sigma) and clarified via centrifugation at 19,650 × g for 20 min at 4°C. Clarified lysates were incubated with equilibrated M2 magnetic flag beads (Sigma-Aldrich, M8823) (100 µl of 50% slurry for every ~2.5 g starting worm pellet) overnight at 4°C. Beads were washed twice with 50 mM HEPES pH 7.6, 150 mM KCl, 10% glycerol and once with TBS (Tris pH 7.6 and 150 mM NaCl). Protein samples were eluted with 150 ng/µl 3xflag peptide (Sigma-Aldrich, F4799) in TBS for 30 min at 4°C with rotation.

In S2A Fig, worm samples were prepared as described above. Briefly, worms were grown at room temperature in an OP50 *E. coli* culture mixed with chicken eggs. Worms were collected and washed in an ice-cold H150 buffer (as described above). Worms were resuspended 1:1 (W/V) in 100 mM HEPES (pH 7.6), 300 mM KCl, 20% glycerol, and 0.1% IGEPAL, with protease inhibitors (Roche, A32965, 1 tablet per 10 mL lysis buffer) and frozen in liquid nitrogen as small pellets. Frozen worm samples were lysed using a coffee grinder pre-chilled with liquid nitrogen until no intact worms were visible under a dissection microscope (Nikon SMZ800N). Lysates were diluted 5-fold with 50 mM HEPES (pH 7.6), 150 mM KCl, 10% glycerol, 0.005% IGEPAL and clarified via centrifugation at 4,000 × g for 10 min at 4°C. Clarified lysates were incubated with equilibrated Anti-flag M2 Magnetic Beads (Sigma-Aldrich, M8823; 100 µl of 50% slurry for every ~2.5 g starting worm pellet) overnight at 4°C. Beads were washed three times with 50 mM HEPES (pH 7.6), 150 mM KCl, 10% glycerol and once with TBS (Tris HCl (pH 7.6) and 150 mM NaCl). Protein samples were eluted with 150 ng/µL 3xflag peptide (Sigma-Aldrich, F4799) in TBS for 30 min at 4°C with rotation.

## MudPIT analysis

Samples were prepared for MudPIT proteomics as described in [25]. Briefly, the Tris concentration of samples was adjusted to 100 mM, tris(2-carboxyethyl)phosphine (TCEP-HCl) (Pierce) was added to 10 mM, and samples were

incubated at 55°C for 1 hr. 2-chloroacetamide (Sigma-Aldrich) was added to 25 mM and samples incubated for 30 min at room temperature, protected from light. Six volumes of pre-chilled acetone were added and precipitation proceeded overnight at -20 °C. Samples were then centrifuged at 8000 x g for 10 min at 4°C. The acetone was aspirated and protein pellets dried. Subsequent protein preparation and downstream analysis were performed exactly as in [25]. Raw data and search results will be deposited to the Proteome Xchange via the MassIVE repository at the time of publication. Mass Spectrometry data will also be accessible at the Stowers Original Data Repository at the time of publication.

## Tissue culture maintenance, plasmid generation, and pulldowns

HEK293T cells were cultured at 37°C with 5% $CO_2$ in 150 mm petri dishes with mammalian growth media (DMEM, 10% Fetal Bovine Serum, 100 U/mL Penicillin, 100 µg/mL Streptomycin, 10 mM HEPES).

Mammalian vectors expressing HaloTag and HA-tagged proteins were generated via Gibson cloning [55]. The control plasmid (pDTW117) contains a linker sequence followed by an HA tag, second linker, TEV cleavage site, and a HaloTag. cDNAs of the *H. sapiens* wild type iASPP (accession number: NP_001135974.1) C-terminus (pDTW123) and mutant iASPP C-termini (pDTW124, and pDTW125) were then cloned into the control plasmid upstream of the initial linker sequence as described in [25]. The HA-tagged PP1 plasmid (pDTW131) was constructed via a one-piece Gibson assembly using an amplicon that was obtained from a plasmid containing *M. musculus* PPP1CA (accession number: NP_114074.1) with oDTW457 and oDTW458. HA-tagged iASPP C-terminus (pDTW231) was constructed via a one-piece Gibson assembly using an amplicon that was obtained from the HaloTagged iASPP C-terminus (pDTW81) with oDTW1113 and oDTW1114. The HA-tagged iASPP C-terminus containing the N657K missense mutation (pDTW232) was made via a one-piece Gibson assembly using an amplicon obtained from the N657K mutant iASPP C-terminus plasmid (pDTW94) with oDTW1113 and oDTW1114. The HA-tagged iASPP C-terminus containing the H665Y missense mutation (pDTW233) was constructed via a one-piece Gibson assembly using an amplicon that was obtained from the H665Y mutant iASPP C-terminus plasmid (pDTW87) with oDTW1113 and oDTW1114.

HEK293T cell transfections and pulldowns were conducted as described in [25]. Briefly, 150 mm dishes of HEK293T cells (~65% confluent) were transfected with a total of 10 µg of plasmid using 60 µg of linear Polyethylenimine 25 kDa (23966, Polysciences Inc.) mixed in 2 ml Opti-MEM (31985070, ThermoFisher). For cotransfection experiments, 5 µg of each plasmid was cotransfected for a total of 10 µg. Approximately 24 hr post-transfection, cells were pelleted and frozen at −80°C. Cell pellets were lysed, clarified, and bound to HaloLink Magnetic Beads (Promega) according to the manufacturer's instructions. Magnetic beads were washed in TBS with 0.09% IGEPAL (CA-630, Sigma). Samples were cleaved (2 h at 21°C with 1,100 RPM shaking) with AcTEV protease (20 units in 100 µl, Invitrogen) to release baits and any bound proteins.

## Western blotting

All SDS-PAGE experiments were run on precast polyacrylamide gels (4–12% Bis-Tris, Invitrogen). Samples were denatured at 95°C for 1 min in 1x Bolt LDS Sample Buffer (ThermoFisher Scientific; B0007) containing 50 mM dithiothreitol. Protein was transferred to PVDF membranes (MilliporeSigma; IPFL85R) using the Pierce Power Blot Cassette system (ThermoFisher Scientific). Blocking and antibody incubation were conducted in Intercept (TBS) Blocking Buffer (Fig 2B, LICOR; 927–60001) or EveryBlot Blocking Buffer (Fig 3B, BioRad, 12010020). Blots were washed using TBS containing 0.1% Tween-20 (v/v) and imaged using the Bio-Rad ChemiDoc MP Imaging System. Band intensities were quantified using the accompanying Image Lab software.

Primary antibodies and dilutions included anti-GFP (1:1,000; Roche, 11814460001, clones 7.1 and 13.1), anti-PP1 (1:1,000; Santa Cruz; Clone E-9; sc-7482), anti-PPP1CA (phospho T320) (1:1,000; Abcam; Clone EP1512Y; ab62334), and anti-HA::Peroxidase (1:500; Roche; clone 3F10; 1201381900). Secondary antibodies and dilutions included anti-mouse HRP antibody (1:20,000; Bio-Rad; 1706516), the goat anti-mouse IRDye 800CW (1:20,000; LICOR; 926–32210)

and the goat anti-mouse IgG (H + L)-HRP Conjugate (1:20,000; Bio-Rad;1706516). Chemiluminescent images were taken following development using the SuperSignal West Dura Extended Duration Substrate (Thermo Scientific; 34075).

## HaloTag gel imaging

In S2A Fig, 22.5 µL of the elution samples were labeled with 1 µL JFX646 dye diluted 1:1000 in TBS for 15 min in the dark. Samples were heated at 95°C for 10 minutes then loaded and separated on an SDS-PAGE gel (Invitrogen, NW04122BOX). The gel was washed in RO water and imaged using the Bio-Rad ChemiDoc MP Imaging System.

## Fluorescent size exclusion chromatography

HEK293T cells were transfected and collected as described in [25]. Cell pellets were thawed in 300 µL Mammalian Lysis Buffer (PAG9381, Promega) with 6 µL 50x Protease Inhibitor Cocktail (G6521, Promega) and sheared using a 25 gauge needle. JFX549 (gifted from Luke Lavis, Janelia Research Campus) was resuspended to 200 µM in DMSO and then diluted 1:40 in TBS. Lysates were labeled for 1 hr on ice in the dark with 1 µL of diluted JFX549. 1 mg/mL ovalbumin, 1 mg/mL aldolase, and 1 mg/mL conalbumin were prepared from Cytiva's Gel Filtration High Molecular Weight kit (28403842, Cytiva). All samples were clarified by centrifugation at 17,000 x g for 5 min at 4°C. Clarified samples were further spun in a Beckman Optima TLX Ultracentrifuge at 200,312 x g for 23 min at 4°C. 100 µL of the resulting supernatants was injected on a Superdex 200 Increase 10/300 GL column using the Shimadzu SIL-20AC HT Prominence Autosampler. Samples were eluted in filter-sterilized (0.22 µm) and degassed TBS for 75 min at 4°C at a rate of 0.5 mL/min using the Shimadzu LC-20AD Prominence Liquid Chromatograph. Fluorescence in elutions was monitored using the Shimadzu RF-20A XS Prominence Fluorescence Detector. General protein was detected using Ex = 288 nm, Em = 350 nm. JFX549 labeled protein was detected using Ex = 549 nm, Em = 571 nm.

## Recombinant protein plasmid generation and purification

The expression plasmid for the iASPP C-terminus (pDTW154) contains the same *H. sapiens* iASPP(602–828) cDNA as described above (Tissue Culture Maintenance, Plasmid Generation, and Pulldowns) followed by an HA tag, HaloTag, HRV-3C protease site, and 6xHis. pDTW154 was constructed by a one-piece Gibson assembly using an amplicon from a plasmid expressing the full-length iASPP cDNA in a pET21b vector (pDTW151). The amplicon was produced using oEP324 and oDTW310. The expression plasmid for PP1 (pDTW258) contains a 6xHis, HRV-3C protease site, and the same *M. mus* PPP1CA cDNA as described above (Tissue Culture Maintenance, Plasmid Generation, and Pulldowns). pDTW258 was constructed by a two-piece Gibson assembly using an amplicon of PPP1CA amplified from a plasmid containing PPP1CA (pDTW102) via oDTW900 and oDTW901. The pET21b vector backbone was amplified from an in-house expression plasmid (pDTW3) using oDTW898 and oDTW899.

BL21(DE3) Competent *E. coli* (NEB, C2527H) cultures transformed with iASPP C-terminus and PP1 expression plasmids were grown at 37°C with agitation (160 rpm) until they reached an $OD_{600}$ of ~1.1. Expression was induced with 100 µM IPTG and cultures were incubated overnight at 18°C with agitation (160 rpm). Cells were collected by centrifugation and stored at -80°C until purification.

A 5 g pellet of bacterial cells was thawed in 50 mM Tris HCl (pH 7.5), 750 mM NaCl, 1 mM 2-mercaptoethanol, Pierce Protease Inhibitor Tablet (1 tablet for every 5 g pellet, ThermoFisher, A32965), 5% glycerol, and 0.15 mg/mL Lysozyme from chicken egg white (Sigma-Aldrich, L6876). The cell slurry was brought to 2.5 mM $MgCl_2$ and 0.5 mM $CaCl_2$ and 60 µL of DNase I (Roche, 10104159001) was then added. PMSF (Sigma-Aldrich, 93482) was added to 0.1 mM. The cell slurry was sonicated while in an ice bath using a Branson Flat Tip for 1/2" Tapped Horns (Amazon, B00DV7NECK) powered by a Sonic Dismembrator Model 500 (Fisher Scientific). The slurry was sonicated at 45% power with 2 minute cycles using 10 sec pulses at 15 sec intervals. The temperature was monitored and maintained between 2–10°C. Sonication continued until the A280 of the clarified lysate no longer increased.

The cell lysate was balanced in high speed 50 mL tubes (Chemglass, CLS-4303-G50) and centrifuged at 19,000 x g for 10 minutes. The supernatant was then filtered through a 0.45 μm filter to make the clarified lysate. 1 mL of 50% TALON Metal Affinity resin (Takara, 635503) was equilibrated using 50 mM Tris HCl (pH 7.5), 750 mM NaCl, 1 mM 2-mercaptoethanol, and 5% glycerol. The resin was mixed with the clarified lysate and the resulting binding reaction was allowed to proceed for two hours at 4°C with end-over-end rotation. The resin was filtered using a 10 mL Econo-Pac Disposable Chromatography Column (Bio-Rad, 732–1010). The resin was washed with 100 column volumes of 50 mM Tris HCl (pH 7.5), 750 mM NaCl, 1 mM 2-mercaptoethanol, and 5% glycerol. The resin was then washed with 50 column volumes of 50 mM Tris HCl (pH 7.5) and 150 mM NaCl. The iASPP C-terminus was eluted from the resin using 50 mM Tris HCl (pH 7.5), 150 mM NaCl, and 150 mM imidazole. PP1 was eluted from the resin by HRV-3C protease cleavage overnight at 4°C with end-over-end rotation in 2 mL of 50 mM Tris HCl (pH 7.5), 150 mM NaCl, and 50 μg/mL HRV-3C protease (purified in-house).

Proteins were further purified by ion exchange using a 5 mL HiTrap Q HP column (GE Healthcare, 17-1154-01) on a Biologic LP purification system and BioFrac Fraction Collector. Protein samples from the TALON affinity purification were diluted to 50 mM Tris HCl (pH 8) and 20 mM NaCl and loaded on the anion column. The protein was then eluted over 20–30 column volumes using a NaCl gradient from 20 mM to 1 M. Protein-containing fractions were pooled and concentrated to the desired level using a 10 kDa Amicon Ultra Centrifugal Filter (Sigma Aldrich, UFC901024) in 50 mM Tris HCl (pH 7.5) and 150 mM NaCl.

## Size exclusion chromatography

Recombinantly purified iASPP C-terminus and PP1 were mixed together at ~1:3 molar ratio, respectively in 50 mM Tris HCl (pH 7.5) and 150 mM NaCl. The protein solution was incubated at 30°C for 1 hr to allow for complex formation. 0.3 mg/mL ovalbumin, 0.3 mg/mL aldolase, and 0.3 mg/mL conalbumin were prepared from Cytiva's Gel Filtration High Molecular Weight kit (28403842, Cytiva). All protein solutions were centrifuged at 200,312 x g as described in the fSEC methods section. 500 μL of each protein solution was injected onto a Superdex 200 Increase 10/300 GL column using an AKTA-purifier system. The column was washed with 30 mL of 50 mM Tris HCl (pH 7.5) and 150 mM NaCl at 0.4 mL/min. Elutions were collected in 0.25 mL fractions using the Frac-950 unit. Fractions from the SEC were run on SDS-PAGE and evaluated using the Lonza SYPRO Ruby Stain protocol (Fisher Scientific, BMA50562).

## Single-molecule TIRF microscopy

Two NGM plates per worm strain were maintained daily at 15°C. The night before an imaging experiment, 200 μL of M9 buffer containing 1.5 μM JF649 (GA1121, Promega) was added dropwise to each plate. Plates of stained worms were wrapped in aluminum foil and stored upright at 15°C overnight. Worms were washed off of each plate using 1 mL 50 mM HEPES pH 7.6, 150 mM KCl and spun at 100 x g for 1 min at 4°C. Worm pellets from two plates were transferred to a new tube and spun at 100 x g for 1 min at 4°C. Excess buffer was removed and worm pellets were resuspended in 30 μL of 50 mM HEPES pH 7.6, 150 mM KCl, 10% (v/v) glycerol, Pierce Protease Inhibitor Tablet (1 tablet for every 12.5 mL buffer, ThermoFisher, A32965) and frozen at -80°C. Worms were thawed on ice and sonicated with a Branson Digital Sonifier using a cup-horn adaptor with 70% power for ~10 min with 3 sec on/off at 4°C. Sonication was repeated until intact worms were no longer visible under a dissection microscope. 30 μL of wash buffer (50 mM HEPES pH 7.6, 150 mM KCl, 10% (v/v) glycerol) was added to each sample before centrifugation at 17,000 x g for 5 min at 4°C. The supernatant was collected and loaded into single-molecule multichamber devices for imaging on a TIRF microscope (described below).

24 mm x 50 mm No. 1.5 glass coverslips (Corning) were marked on one corner with a diamond stylist and placed in a glass holder. Coverslips were treated for 20 min in a Harrick Plasma Plasma Cleaner (PDC-001) on the "high" setting with a pressure between 0.4-0.8 Torr, established with a IDP-3 Dry Scroll Pump and Harrick Plasma Vacuum Gauge (PDC-VCG). Cleaned coverslips were then PEGylated by applying 45 μL of a 1:100 mixture of 1% biotin-PEG-silane in ethanol

(Laysan Bio Biotin-PEG-SIL-2K-1g) and PEG-silane (85%, VWR 77035–498) between two coverslips such that their marked sides are in contact with the PEGylation solution. Coverslips incubated at room temperature in the dark for 1 hr. Coverslips were washed thoroughly in RO water and dried with nitrogen gas before storage at room temperature in the dark in a glass Tupperware container with Drierite desiccant (VWR) until use.

Multichamber devices were prepared as in [40]. Briefly, multichamber devices of ~30 μL volumes were fashioned by adhering PEGylated coverslips to glass slides using ~4 mm strips of double-sided adhesive tape orthogonal to the long axis. Chambers were equilibrated with two volumes of wash buffer by capillary action. Chambers were washed twice after each of the following steps: 10 min incubation with 0.2 mg/mL neutravidin, 10 min incubation with biotinylated flag antibody (F9291, Sigma Aldrich) diluted 1:100 in wash buffer, and 10 min incubation with blocking buffer (50 mM HEPES pH 7.6, 150 mM KCl, 10% (v/v) glycerol, 20 mg/mL BSA). 30 μL of clarified whole worm lysates (as described above) were loaded into chambers and incubated at room temperature for 60–80 min in a dark humidified container prior to imaging. Immediately before imaging, the sample chamber was washed once with 30 μL of wash buffer.

Samples were imaged on a custom-built TIRF microscope (as described above) with the addition of a Zero-order quarter-wave plate (WPQ10M-633, ThorLabs) to circularly polarize the 647 nm laser (Coherent OBIS). Samples were imaged continuously at 20 frames/sec for ~50 sec while simultaneously exciting with 488 nm (~4 mW) and 647 nm (~21 mW) lasers for a total of 1000 frames/channel. Laser powers were tuned such that the bleach half-time of our JF646 labeled HaloTag::GSP-2 was ~1/5 of our total imaging time and our APE-1::GFP::3xflag was ~1/10 of our total imaging time. Bleach half-times were calculated using a half-time calculation software at https://github.com/dickinson-lab/SiMPull-Analysis-Software/blob/master/Static_Analysis/Visualization/bleachHalfTime.m [56]. Each single-molecule imaging session began by acquiring a registration image, in which 0.1 μm TetraSpeck fluorescent microsphere beads (T7279, ThermoFisher) were imaged in green and far-red channels, to allow for proper colocalization analysis in subsequent steps.

Training data for our CNNs was collected across 50 technical replicates for each of two biological replicates of the strain expressing APE-1::GFP::3xflag; HaloTag::GSP. We used a modified version of the analyze_batch.m function at https://github.com/dickinson-lab/SiMPull-Analysis-Software/blob/master/Static_Analysis [56], in which the automated step-counting feature is disabled and the "maxSpots" variable is set equal to "imgArea/ 3e5." We also modified the coloc_spot.m utility function by setting the "colocDistance" variable equal to 2 pixels. These modified functions were used to identify and catalog colocalized green and far-red spots along with their corresponding fluorescence intensity traces.

We then employed custom software to randomly display either green or far-red Z-score normalized fluorescence intensity traces from colocalized spots. Traces were manually labeled based on the number of discrete photobleaching events – defined as stepwise, sustained (>25 counts) decreases in fluorescence. Traces were labeled as having "at least x" steps, where x is the value of photobleaching events across the entire trace or until early stopping. Counting stopped early when the intensity trace increased by a step.

Far-red traces were binned into four classes: rejected, 1-step, 2-step, or 3-and-higher-step. Far-red traces were labeled as rejected if no steps were counted or if the signal variability prevented visualization of discrete drops in fluorescence. We combined 3-step with higher steps traces into one class to simplify CNN-training. Green traces were binned into two classes: rejected or 1-step. Green traces were labeled as rejected if no steps were counted, if the signal variability prevented visualization of discrete drops in fluorescence, or if there was more than 1 photobleaching event in the trace.

In total, 30,135 far-red traces from both biological replicates were labeled: 12,563 rejected, 10,926 1-step, 5,053 2-step, and 1,593 3-and-higher-step. 13,773 green traces from one biological replicate were labeled: 4,329 rejected and 9,444 1-step. To account for class imbalances, we applied Inverse Class Frequency Weighting, which assigns greater weight to underrepresented classes when calculating loss values during training.

CNN training for each fluorescent channel began with an exploratory phase in which we tested 39 distinct model architectures. Each model architecture tested different kernel sizes, number of 1D filters, or the configuration of fully connected (dense) layers. For each model architecture, we tested various learning rates and batch sizes to optimize the model

training protocol. Each configuration used a stratified, randomized 80/20 train-validation split of the respective training dataset. We selected the top five configurations based on validation accuracy. Final configuration selection for each channel was made using 5-fold cross validation, evaluating average accuracy and F1 scores across folds. Then, the final CNN-FarRed and CNN-Green models were trained on 100% of their respective labeled datasets using the selected architecture and hyperparameter configuration.

9-10 biological replicates, each consisting of 7 technical replicates, were collected for each experimental sample. Colocalized far-red and green spots were identified using the modified analyze_batch.m function described above. We used custom Python scripts to perform the following actions: 1) extract and Z-score normalize far-red traces from colocalized spots; 2) use the CNN-FarRed to assign far-red traces into appropriate classes; 3) extract and Z-score normalize green traces from colocalized spots; 4) use the CNN-Green to assign green traces into appropriate classes; 5) filter data such that only cases where a non-rejected green trace accompanied by its respective non-rejected far-red trace remain for final analysis. The total number of traces for each class of far-red traces was summed. The percentage of traces in each class for each replicate was calculated.

Labeling efficiency was calculated using a strain expressing mNeonGreen::HaloTag under the *mex-5* promoter. Worms were stained, collected, and lysed as described above. An unstained control plate was included in each biological replicate. Samples were diluted 1:10 in 50 mM HEPES (pH 7.6), 150 mM KCl, 10% (v/v) glycerol just prior to loading into single-molecule, multichamber devices. Chambers were washed and imaged as described above. Three biological replicates, each consisting of 10 technical replicates, were collected for the stained and unstained samples. The percentage of mNeonGreen spots colocalized with HaloTag spots was quantified using https://github.com/dickinson-lab/SiMPull-Analysis-Software/blob/master/Static_Analysis [54], which contained the coding changes described above.

## Statistical analysis

Localization data of APE-1 to epithelial junctions was analyzed in GraphPad Prism using one-way ANOVA with Tukey's adjustment for multiple comparison. Jowls assays were analyzed in RStudio as previously described in [25]. Briefly, a generalized linear model was fit to a binomial distribution via RStudio for jowls assays. Body length assays were analyzed using one-way ANOVAs with Tukey's adjustment for multiple comparisons. Each CNN was evaluated using a custom python script to perform 5-fold cross validation. Single-molecule photobleaching data were analyzed in GraphPad Prism using a nested one-way ANOVA with Tukey's adjustment for multiple comparisons.

## Supporting information

**S1 Fig. APE-1 function requires its N-terminal helix.** (A) Alphafold structural domain predictions mapped onto the primary sequences of ASPP1, ASPP2, and iASPP from H. sapiens as well as APE-1 from C. elegans. Predicted structures of the N-terminal alpha helices are shown in the insets above each domain map with predicted residues numbers indicated. The predicted structures are colored according to each residue's pLDDT score provided by Alphafold. Represented in different colors are the beta-grasp domain (light blue), N-terminal alpha helix (tan), undefined alpha-helical regions (brown), ankyrin repeats (light green), and SH3 domain (teal). (B) Body length assay. Data represent mean and S.E.M. (black bars) of 23–29 biological replicates. (C) MLT-4::RFP junctional localization assay of images in Fig 1B. Data represent the mean and S.E.M. (black bars) of 10 biological replicates. p values indicated above comparison brackets.
(TIF)

**S2 Fig. Testing ASPP-dependency of phosphatase oligomerization.** (A) 3xflag immunoprecipitations from worms expressing P*semo-1*::3xflag::GFP baits and an endogenous HaloTag(HT)::GSP-2 prey. Prey is labeled with JFX646 dye (top) and bait is immunostained with anti-GFP (middle). HT intensity is normalized to the first lane (bottom). (B) Jowls assay of CCDC85 and RASSF8 knockouts (two independent alleles each). Data represent percent jowls and 95%

confidence interval (black and gray bars) in adult animals (n = 40). (C) Size exclusion chromatogram of recombinantly purified iASPP(C)::HT (black line) and iASPP(C)::HT mixed with PP1 at a 1:3 molar ratio (red line; top). SYPRO Ruby-stained denaturing gels of indicated elution fractions (bottom). iASPP(C)::HT is ~62.7 kDA and PP1 is ~37.9 kDa. Peaks for molecular weight standards (described in methods) are represented as vertical black lines with the top 10% of signal shaded in light gray.
(TIF)

**S3 Fig. Convolutional Neural Network (CNN) confusion matrices.** (A) CNN-Green mean confusion matrix of raw counts averaged across 5-fold cross validation during training with GFP fluorescence intensity traces. (B) Row-normalized mean confusion matrix from panel A showing recall for each class during K-fold cross validation of the CNN-Green. (C) CNN-FarRed mean confusion matrix of raw counts averaged across 5-fold cross validation during training with far-red fluorescence intensity traces. (D) Row-normalized mean confusion matrix from panel C showing recall for each class during K-fold cross validation of the CNN-FarRed. (E) Expanded view of data from Fig 4B. p values indicated above comparison brackets. Data represent mean and S.E.M. (black bars). (F) HaloTag labeling efficiency *in vivo*. Animals broadly expressing mNeonGreen::HaloTag under the *mex-5* promoter were fed JF646 dye. Whole animal lysates were bound to functionalized coverslips via anti-mNeonGreen and imaged using TIRF microscopy. The percentage of mNeonGreen spots colocalized with far-red JF646 spots was quantified. Unlabeled animal lysates were used as a negative control. Data represent mean and S.E.M. (black bars) of three biological replicates each containing ten technical replicates.
(TIF)

**S4 Fig. GSP-2 oligomers bypass APE-1 missense mutations.** (A) Body length assay of animals with endogenous GSP-2 oligomerized via EODs. Data represent the mean and S.E.M. (black bars) of 27–36 biological replicates. **** indicates $p < 0.0001$. * indicates $p < 0.05$. (B) Jowls assay of animals expressing a skin-specific, single-copy insert of GSP-2 or Cry2olig(Cry2)::GSP-2. Data represent percent jowls and 95% confidence interval (black and gray bars) in adult animals (n = 25–35 each). (C) Jowls assay of animals expressing skin-specific Cry2::GSP-2 in MLT-4::RFP; APE-1(N583K) mutants. Animals were grown at 24°C or 15°C under light or dark conditions. Data represent percent jowls and 95% confidence interval (black and gray bars) in adult animals (n = 29–46 each).
(TIF)

**S1 File. Training metrics from CNN-Green K-fold cross validation.**
(XLSX)

**S2 File. Training metrics from CNN-FarRed K-fold cross validation.**
(XLSX)

**S3 File. Strains, alleles, and reagents.**
(DOCX)

## Acknowledgments

We thank the labs of Toshi Kawate, Maurine Linder, Carrie Adler, Carolyn Sevier, Richa Sardana, Josh Chappie, and Natasza Kurpios for sharing space, equipment, and reagents. We thank Ed Partlow III, Scott D. Emr and Chris Fromme for valuable advice throughout the project. For technical advice and assistance, we thank Wendy Greentree (tissue culture), Sarah Chan and Katie Crowther (SEC), and Jacqueline Ehrlich (fSEC). We thank Dan J. Dickinson, Eric Drier, and Jeff Lange for guidance and advice in our single-molecule TIRF microscopy. We also thank Dan J. Dickinson for sharing plasmids containing the EODs. We thank Luke Lavis for sharing various fluorescent dyes. We thank Matt Thomas at the Cornell Statistical Consulting Unit for advice on statistical analyses. ChimeraX was developed by the Resource for

Biocomputing, Visualization, and Informatics at the University of California, San Francisco, with support from National Institutes of Health R01-GM129325 and the Office of Cyber Infrastructure and Computational Biology, National Institute of Allergy and Infectious Diseases.

## Author contributions

**Conceptualization:** Derek T. Wei, Erika Beyrent, Laurence Florens, Gunther Hollopeter.

**Data curation:** Derek T. Wei, Kayleigh N. Morrison, Gwendolyn M. Beacham, Cyrus A. Habas, Ying Zhang, Laurence Florens, Gunther Hollopeter.

**Formal analysis:** Derek T. Wei, Kayleigh N. Morrison, Cyrus A. Habas, Laurence Florens.

**Funding acquisition:** Gunther Hollopeter.

**Investigation:** Derek T. Wei, Kayleigh N. Morrison, Gwendolyn M. Beacham, Cyrus A. Habas, Ying Zhang.

**Methodology:** Derek T. Wei, Gwendolyn M. Beacham, Erika Beyrent, Ying Zhang, Laurence Florens, Gunther Hollopeter.

**Project administration:** Derek T. Wei, Laurence Florens, Gunther Hollopeter.

**Resources:** Derek T. Wei, Gwendolyn M. Beacham, Erika Beyrent, Laurence Florens, Gunther Hollopeter.

**Software:** Derek T. Wei, Erika Beyrent, Ying Zhang.

**Supervision:** Derek T. Wei, Laurence Florens, Gunther Hollopeter.

**Validation:** Derek T. Wei, Kayleigh N. Morrison, Gwendolyn M. Beacham, Cyrus A. Habas, Laurence Florens, Gunther Hollopeter.

**Visualization:** Derek T. Wei, Kayleigh N. Morrison, Erika Beyrent, Cyrus A. Habas, Gunther Hollopeter.

**Writing – original draft:** Derek T. Wei, Kayleigh N. Morrison.

**Writing – review & editing:** Derek T. Wei, Kayleigh N. Morrison, Gwendolyn M. Beacham, Erika Beyrent, Cyrus A. Habas, Ying Zhang, Laurence Florens, Gunther Hollopeter.

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
