## [Decision Letter · Decision Letter 0]

29 Jun 2025

PGENETICS-D-25-00589

ASPPs multimerize protein phosphatase 1

PLOS Genetics

Dear Dr. Hollopeter,

Thank you for submitting your manuscript to PLOS Genetics. After careful consideration and evaluation by three reviewers, we feel that it has merit but does not fully meet PLOS Genetics's publication criteria as it currently stands. Therefore, we invite you to submit a revised version of the manuscript that addresses the points raised during the review process. In particular, we feel that it is important to address comments raised by Reviewer 3 on whether other proteins associated with ASPP/PP1 could aid in oligomerization of PP1. 

Please submit your revised manuscript within 60 days Aug 28 2025 11:59PM. If you will need more time than this to complete your revisions, please reply to this message or contact the journal office at plosgenetics@plos.org. Please include the following items when submitting your revised manuscript:

We look forward to receiving your revised manuscript.

Kind regards,

Jeremy Nance

Academic Editor

PLOS Genetics

Fengwei Yu

Section Editor

PLOS Genetics

Aimée Dudley

Editor-in-Chief

PLOS Genetics

Anne Goriely

Editor-in-Chief

PLOS Genetics

**Journal Requirements:**

At this stage, the following Authors/Authors require contributions: Derek T. Wei, Kayleigh N. Morrison, Gwendolyn M. Beacham, Erika Beyrent, Ying Zhang, Laurence Florens, and Gunther Hollopeter. Please ensure that the full contributions of each author are acknowledged in the "Add/Edit/Remove Authors" section of our submission form.

The list of CRediT author contributions may be found here: https://journals.plos.org/plosgenetics/s/authorship#loc-author-contributions

- ® on pages: 16, and 21

- TM on pages: 21, and 22.

Potential Copyright Issues:

i) Figures 2B, 3, and 4A. Please confirm whether you drew the images / clip-art within the figure panels by hand. If you did not draw the images, please provide (a) a link to the source of the images or icons and their license / terms of use; or (b) written permission from the copyright holder to publish the images or icons under our CC BY 4.0 license. Alternatively, you may replace the images with open source alternatives. See these open source resources you may use to replace images / clip-art:

5) When completing the data availability statement of the submission form, you indicated that you will make your data available on acceptance. We strongly recommend all authors decide on a data sharing plan before acceptance, as the process can be lengthy and hold up publication timelines. Please note that, though access restrictions are acceptable now, your entire data will need to be made freely accessible if your manuscript is accepted for publication. This policy applies to all data except where public deposition would breach compliance with the protocol approved by your research ethics board. If you are unable to adhere to our open data policy, please kindly revise your statement to explain your reasoning and we will seek the editor's input on an exemption. Please be assured that, once you have provided your new statement, the assessment of your exemption will not hold up the peer review process.

6) Please ensure that the funders and grant numbers match between the Financial Disclosure field and the Funding Information tab in your submission form. Note that the funders must be provided in the same order in both places as well. Currently, this grant "R01GM127548" is missing from the Funding Information tab.

7) Thank you for indicating that "The authors declare no competing interests." Please state "The authors have declared that no competing interests exist".

**Reviewers' comments:**

Reviewer's Responses to Questions

Reviewer #1: The manuscript by Wei and colleagues is a well-focused study demonstrating that ASPP proteins likely act by oligomerizing PP1 (Ser/Thr phosphatase) enzymes. In addition, The C. elegans N-terminus of APE-1 is required for its localization to apical junctions located between hyp7 and the seam cell of the worm epidermis. A major strength of the study are the multiple complementary methods used to show that ASPP proteins promote PP1 oligomerization through a domain in the C-terminus and that this impacts the function of APE-1/ASPP–GSP-2/PP1 based on functional readouts in the worm. Moreover, induced oligomerization of GSP-2 bypasses defects in the C-terminus of APE-1, demonstrating that oligomerization is really a crucial component of the function of APE-1 in addition to localizing PP1 to apical junctions. The paper is concise and well written with carefully drawn conclusions and should be of considerable interest to several fields. I have no requested experiments, and my relatively few comments and questions may be viewed as suggestions.

1) Fig 1. The signal in the representative image for MLT-4::RFP–H appeared a good bit dimmer than the other images. Was this a trend and, if so, does it suggest that expression of APE-1–H somehow affects the ability of MLT-4 to associate with these junctions?

2) Fig S1. I was curious what the pLDDT view of the helices in S1 Fig might look like. My understanding is that AF likes to create helices that are very low confidence even when regions are potentially unstructured. So, I was curious about the confidence of those predictions and if there is any experimental support for helices, such as crystal structures, which were mentioned in the Discussion.

3) Fig 2. Consider adding some basic information about the nature of the two iASPP-C mutations to the figure legend, as these are not currently mentioned.

4) Fig 3. In 3C it isn’t immediately clear what the label Halotag indicates since all the constructs should have the tag. Perhaps list it as “control” and provide an estimated molecular weight, as this is not stated in the methods.

5) Fig 4. Although the data in 4B clearly indicate that the proportion of oligomeric (2 or more) complexes decreases with the N583K mutation, it does so by less than one might expect given some of the other data. In addition, the confusion matrices in S4 suggest a decent ability to distinguish between 2 and 3 steps. So, it was unclear why these weren’t compared among the genotypes. For example, did H591Y show any distinction between 2 and 3 versus WT? Even if differences were not observed, it could still be informative to show that data (as a sup) and to try and account for why the different techniques, while agreeing generally, didn’t correlate especially well. Admittedly, this was touched upon in the Discussion, but a clear hypothesis for the discrepancies wasn’t forwarded. (Perhaps there is none.)

6) Discussion. Connected point #5, might the defect in H591Y have more to do with the orientation of the oligomerized PP1 subunits, such that while forming oligomers they are still inactive? By chance, were any other kinds of EODs tested that failed to work? Also, what was the origin of the EOD used in this study and why was it chosen?

As for all my reviews this is signed by David Fay.

Reviewer #2: Protein phosphorylation is a critical post-translational modification that influences cell signaling and behavior. Regulation of the protein kinases that catalyze phosphorylation has been extensively studied, but regulation of phosphatases, which reverse this modification, is comparatively underexplored. Here, Wei and colleagues have used genetic and biochemical experiments to study regulation of protein phosphatase 1 (PP1) by ASPP proteins and their C. elegans homolog, APE-1. Their data show that ASPP recruits PP1 into clusters, and this clustering is necessary and sufficient for normal PP1 activity in C. elegans.

This a really nice, straightforward paper that integrates complementary approaches to reach an interesting result. Indeed, the manuscript was so easy to read that while reading, it was tempting to forget what tremendous amount of work must have gone into combining engineered loss- and gain-of-function experiments, structure-function approaches, and both bulk and single-molecule biochemistry into a such a cohesive story. Well done. I have no major concerns about the manuscript and only a few minor points for the authors to consider:

- It would be nice if the authors could be a bit more precise about the molecular weights they expect in their fSEC experiments. They write “A 1:1 ratio [of iASPP to PP1] would be ~100 kDa.” According to Uniprot, iASPP is 89 kD and PP1 is 36 kD, so a 1:1 ratio would be 125 kD. Also, if possible, it would be good to provide a control for how unbound, monomeric iASPP runs on their column – perhaps by doing the separation under denaturing conditions or by starting with cells depleted of PP1.

- The fSEC and SiMPull measurements show the same trend, which is nice, but I was surprised not to see more larger clusters in the SiMPull experiments. How efficient is HaloTag labeling on worm plates, and could under-counting of HaloTags due to incomplete labeling explain why there aren’t more larger clusters? If the authors don’t have an estimate of labeling efficiency under their conditions, they can simply acknowledge the possibility that labeling is incomplete as a possible caveat.

- Along the same lines, the authors trained their classifier to distinguish dimers from multimers, but then didn’t report the results of that classification with respect to GSP-2. Were complexes with 3+ GSP-2 subunits ever observed, and if so, how often? It would be good to report this (perhaps in the supplement).

Reviewer #3: In the present manuscript, Wei et al use C. elegans genetics and biophysical techniques to study the role of the scaffold protein APE-1 (ASPP family member iASPP in mammals) in regulating the activity and localisation of the PP1 catalytic subunit. They show that APE-1 associates with cell junctions via its N-terminal helical region and confirm previous findings that ASPP proteins associate with PP1 through their C-terminus. Unexpectedly, the authors uncover evidence that APE-1 nucleates PP1 oligomerisation. They identify loss-of-function mutations in the ASE-1 ankyrin repeats that affect its ability to associate with super-stoichiometric amounts of PP1 using co-immunoprecipitations, gel filtration and a TIRF assay. The data are generally solid and well presented. The ability of a regulatory subunit to promote PP1 oligomerisation is a new observation that adds to our understanding of how phosphatase activity is targeted to the correct subcellular localisation to ensure appropriate substrate dephosphorylation. The study should therefore be of broad interest, however the points below should be addressed in order to strengthen the authors’ conclusion before the manuscript can be accepted in PLoS Genetics.

1) Looking at the published structures of iASPP/PP1 (ref 45) and ASPP2/PP1 (ref 24), it is not clear how several PP1 subunits could associate simultaneously with ASE-1. Could the authors model this possible association using AlphaFold to propose some possible scenarios that might allow this to happen?

2) Along the same lines, although the authors do a good job of using orthogonal approaches to validate their model of PP1 oligomerisation, none of the experimental setups exclude that for instance other proteins associated with the ASPP/PP1 complex (e.g. CCDC85 and RASSF family members in the case of ASPP2/Drosophila ASPP) could participate in PP1 oligomerisation. For example, the gel filtration experiments are consistent with multiple PP1 subunit being present in the slower-migrating complexes, but other proteins could also cause (part of) this shift and be sensitive to the N583K mutation. I would therefore suggest the following experiments:

- Perform the TIRF assay in Figure 4 using the ASE-1 C-terminal construct. This would reduce the likelihood that other ASE-1 partner are participating in oligomerisation.

- Perform the TIRF assay in Figure 4 by tethering a Flag-GFP-tagged PP1 and imaging HALO-tagged PP1. This would test the ability of PP1 itself to oligomerise.

- In an ideal world, the authors would perform biophysical assays/gel filtration and/or GST pulldowns on bacterially expressed and purified ASE-1 C-ter and PP1 to prove the oligomerisation model in a “clean” system, though I realise they may not have these approaches established in their lab.

I should say that if it does prove that for example other proteins are involved in forming PP1 oligomers, this would not preclude publication. These experiments are suggested to help the authors refine their model.

3) Figure 4B: for the readers to better appreciate the difference between the mutant and wt APE-1, it would be good to provide a supplementary graph showing the relative distribution of the different oligomeric forms (ASE-1+1/2/3+ PP1) in mutants vs wt rather than just the percentage of complexes with oligomers as shown in the figure.

**Have all data underlying the figures and results presented in the manuscript been provided?**

Reviewer #1: **No: ** It is unclear to what extent the raw data/numbers are required that contributed to some of the figures. We always put everything into a spreadsheet, but maybe that is more than necessary.

Reviewer #2: Yes

Reviewer #3: Yes

PLOS authors have the option to publish the peer review history of their article (what does this mean?). If published, this will include your full peer review and any attached files.

Reviewer #1: **Yes: ** David S. Fay

Reviewer #2: No

Reviewer #3: No

**Figure resubmission:**
---

## [Decision Letter · Decision Letter 1]

2 Sep 2025

Dear Dr Hollopeter,

We are pleased to inform you that your manuscript entitled "ASPPs multimerize protein phosphatase 1" has been editorially accepted for publication in PLOS Genetics. Congratulations!

Yours sincerely,

Jeremy Nance

Academic Editor

PLOS Genetics

Fengwei Yu

Section Editor

PLOS Genetics

Aimée Dudley

Editor-in-Chief

PLOS Genetics

Anne Goriely

Editor-in-Chief

PLOS Genetics

Comments from the reviewers (if applicable):

Reviewer #1:

Reviewer #2:

Reviewer #3:

Reviewer's Responses to Questions

**Comments to the Authors:**

Reviewer #1: The authors did a very thorough job of addressing reviewer comments.

Reviewer #2: I was enthusiastic about the first version of the manuscript, and the authors have fully addressed my (minor) comments. I have no further concerns. Congrats on a nice piece of work.

Reviewer #3: The authors have performed several new experiments to answer my queries. The possibility remains open that proteins other than ASPP participate in PP1 oligomerization (including PP1 itself), which the authors appropriately acknowledge in the text. I therefore support publication of the manuscript in PLoS Genetics.

**Have all data underlying the figures and results presented in the manuscript been provided?**

Reviewer #1: Yes

Reviewer #2: Yes

Reviewer #3: Yes

PLOS authors have the option to publish the peer review history of their article (what does this mean?). If published, this will include your full peer review and any attached files.

Reviewer #1: **Yes: ** David Fay

Reviewer #2: No

Reviewer #3: No

**Data Deposition**

http://datadryad.org/submit?journalID=pgenetics&manu=PGENETICS-D-25-00589R1

**Press Queries**

---

## [Editor Report · Acceptance letter]

PGENETICS-D-25-00589R1

ASPPs multimerize protein phosphatase 1

Dear Dr Hollopeter,

We are pleased to inform you that your manuscript entitled "ASPPs multimerize protein phosphatase 1" has been formally accepted for publication in PLOS Genetics! Your manuscript is now with our production department and you will be notified of the publication date in due course.

With kind regards,

Zsofia Freund

PLOS Genetics

On behalf of:
